# Consumption of Energy Drinks and Attitudes Among School Students Following the Ban on Sales to Minors in Poland

**DOI:** 10.3390/nu17193167

**Published:** 2025-10-08

**Authors:** Regina Ewa Wierzejska, Anna Małgorzata Taraszewska, Agnieszka Wiosetek-Reske, Anna Poznańska

**Affiliations:** 1Department of Nutrition and Nutritional Value of Food, National Institute of Public Health NIH—National Research Institute, Chocimska St. 24, 00-791 Warsaw, Poland; 2Medical Center for Dietetics and Health Education, National Institute of Public Health NIH—National Research Institute, Chocimska St. 24, 00-791 Warsaw, Poland; ataraszewska@pzh.gov.pl; 3Department of Nutrition, Medical University of Applied and Holistic Science, Marcina Kasprzaka St. 49, 01-234 Warsaw, Poland; 4Department of Population Health Monitoring and Analysis, National Institute of Public Health NIH—National Research Institute, Chocimska St. 24, 00-791 Warsaw, Poland

**Keywords:** energy drinks, sales ban, minors, consumption

## Abstract

**Background:** In 2024, Poland introduced a ban on the sale of energy drinks to individuals under 18 years of age. The aim of this study was to assess energy drink consumption among adolescents aged 12–17 years and to evaluate the effectiveness of this regulation. **Methods:** This cross-sectional study was conducted in 2025, using an anonymous questionnaire. A total of 1691 students from primary and secondary schools, living in both urban and rural areas, participated. **Results:** Consumption of energy drinks in the month preceding the survey was reported by 41.1% of students, with the likelihood of consumption increasing by nearly 50% with each additional year of age (OR = 1.496; 95% CI: 1.381–1.621; *p* < 0.001). Secondary school students reported significantly higher consumption compared with primary school students (47.1% vs. 21.6%; *p* < 0.001). The median consumption frequency was 1–2 times per month among primary school students and 1–2 times per week among secondary school students. More than half of adolescents (58.1%) stated that the sales ban did not restrict their access to energy drinks, with adult-mediated purchases being the most common source. Among those attempting direct purchases in physical shops, only 19.3% were consistently asked to provide proof of age. Over half of the respondents did not believe that energy drinks can be harmful to their health; these students reported consumption more than twice as often as students who regarded them as harmful (54.0% vs. 23.9%; *p* < 0.001). **Conclusions:** These findings suggest that, to date, the ban on energy drink sales to minors in Poland has had a limited impact on adolescent consumption, highlighting the need for enhanced educational initiatives in this area. However, the study was cross-sectional in nature and was not conducted on a nationally representative sample of adolescents, which should be taken into account when interpreting the results.

## 1. Introduction

Energy drinks have been available on the Polish market since 1995. Similarly to global trends, they rapidly gained popularity and have become a substantial segment of the non-alcoholic beverage market. In Poland, sales continued to grow even during the COVID-19 pandemic [1], mirroring trends in the United States, where a consistent year-on-year increase has been observed for over a decade [2]. It is worth noting that one of the conditions for introducing energy drinks to the Polish market was the inclusion of a warning stating *‘product intended for adults only’* on the packaging, due to presumed adverse effects on the developing body [3]. Following Poland’s accession to the European Union, manufacturers were required to comply with EU regulations, which require a less stringent label reading *‘not recommended for children’* [4]. Such warnings, typically placed on the back of cans, appear to have little influence on purchasing decisions, as scientific evidence consistently indicates frequent consumption of these drinks by children and adolescents, both in Europe and globally [5,6,7]. Italian studies have shown that first consumption of energy drinks most often occurs before the age of 13 [8,9].

For years, the advertising and marketing of energy drinks have been targeted at young people. In the past, promotional messages even included statements such as *‘this size of can is very comfortable to hold, especially for children’* [10] and the energy drink Cocaine was sold in the United States as *‘The legal alternative’* [11]. Although in 2014 the American Beverage Association established guidelines for responsible labelling and marketing of these drinks, including restrictions on marketing aimed at children, a report on compliance indicated little improvement and most producers continue to target children [12]. Similar patterns have been observed in Canada, where contrary to regulations, advertisements are directed at children and promote energy drinks for use during sports [13,14] and in Australia, where social media platforms such as TikTok frequently depict adolescents consuming these beverages [15].

Scientific evidence consistently emphasises that energy drink consumption by children and adolescents is associated with health risks. Compared with non-consumers or occasional consumers, regular consumers are more likely to experience sleep disturbances, headaches, gastrointestinal problems, hyperactivity and difficulties with learning. Very serious consequences of energy drink consumption may include cardiovascular events, as well as mental health problems such as anxiety, depression and, in some adolescents, even suicidal thoughts [5,6,7,16,17,18,19]. Another concerning finding is that children who consume energy drinks are more likely to initiate alcohol and cigarette use at an earlier age [18,20]. The literature also highlights that the long-term effects of energy drink consumption among adolescents remain insufficiently understood. However, particular concern relates to the potential adverse effects on the central nervous system, which is still in the developmental stage [6,18,19].

Considering the above, an increasing number of countries have undertaken measures to reduce energy drink consumption among children and adolescents. It is estimated that 55 countries worldwide have introduced a tax on these beverages, 33 countries have banned their sale in schools and 23 countries have implemented other restrictions on sales. For example, Sweden prohibits the sale of energy drinks to individuals under 15 years of age, Lithuania restricts sales to those under 18 and in the United Kingdom, supermarkets have voluntarily stopped selling these beverages to individuals under 16 years of age [17,21]. On 1 January 2024, Poland introduced a ban on the sale of energy drinks (defined as beverages containing more than 150 mg/L of caffeine or those with added taurine) to individuals under 18 years of age, which also applies to sales through vending machines [22]. It is also worth noting that earlier, from January 2021, energy drinks, along with all sugar-sweetened and caffeinated beverages, became subject to taxation in Poland [23].

This study sought to assess the extent to which the sales ban has reduced energy drink consumption among adolescents and to examine adolescents’ perceptions of retailers’ compliance with the regulations.

## 2. Materials and Methods

### 2.1. Study Design

This cross-sectional study was conducted between February and June 2025, using an anonymous questionnaire. The questionnaire covered knowledge of the existence of a ban on the sale of energy drinks, the frequency of their consumption and issues related to obtaining these beverages. The understanding of the questionnaire designed by the authors was previously tested on a group of several young people. The English version of the questionnaire is attached as Appendix A. The study included 1691 adolescents aged 12–17 years from 13 schools, comprising 398 students in grades 7–8 of primary school and 1293 students in grades 1–3 of secondary school, after obtaining written parental consent. The recruitment process is illustrated in Figure 1. Participants attended schools located in three Polish provinces: Mazowieckie (Marsovian), Łódzkie (Łódź) and Lubelskie (Lublin), including a large city (Warsaw) and two smaller towns (Łowicz and Łuków). Boys accounted for 58.6% of the sample, girls for 41.3% and 0.1% self-identified as non-binary. A total of 54.6% of students resided in urban areas, while 45.4% lived in rural areas.

### 2.2. Statistical Analysis

The survey results were presented using descriptive statistics. The frequency of responses to categorical questions was expressed as percentages. The percentages do not always add up to 100% due to rounding. Age (quantitative variable) was characterised by specifying the range of variation, quartiles, mean value and standard deviation; the average frequency of drinking beverages (ordinal variable) was presented using the median.

The distributions of responses were compared in subgroups of respondents classified by gender (girls vs. boys, excluding two non-binary respondents), type of school (primary vs. secondary), place of residence (urban vs. rural) and frequency of drinking (drinking 3–4 times a week or more vs. other students). The frequency of responses to qualitative questions was compared using the chi-square test and the age distribution was compared using the Mann–Whitney test. To determine the strength of the association between drinking frequency and age, the odds ratio with a 95% confidence interval was calculated. Since the aim of the study was not to assess cause-and-effect relationships, but rather to provide a general assessment of the situation after the introduction of the sales ban, due to the relatively narrow scope of the data and the nature of the study, we limited ourselves to descriptive statistical methods.

In all statistical tests, a significance level of 0.05 was assumed. The calculations were performed using Stata 16.1.

## 3. Results

A total of 97.3% of students reported being aware of the ban on the sale of energy drinks to minors in Poland, including 96.2% of primary school students and 97.7% of secondary school students. No statistically significant differences were observed according to school type, gender or place of residence.

### 3.1. Energy Drink Consumption Among Adolescents Following the Sales Ban

Consumption of energy drinks in the month preceding the survey was reported by 41.1% of students. Consumption was significantly more frequent among secondary school students (47.1%) compared with primary school students (21.6%) (*p* < 0.001) and among students living in rural areas (50.2%) compared with those in urban areas (33.5%) (*p* < 0.001). No statistically significant difference was observed between boys and girls (41.9% vs. 40.0%). The proportion of students consuming energy drinks increased with age, with each additional year of age associated with an almost 50% higher likelihood of consumption (odds ratio, OR = 1.496; 95% CI: 1.381–1.621; *p* < 0.001).

The largest proportion of students reported consuming energy drinks 1–2 times per month (20.8%), but a notable proportion reported daily or nearly daily consumption (7.3%) (Figure 2).

Frequent consumption of energy drinks, defined as consumption 3–4 times per week or more, was reported by 10.7% of all students. This group was, on average, slightly older than those who consumed energy drinks less frequently or not at all and more often comprised a higher proportion of secondary school students and rural residents (Table 1).

Considering only the group of students who reported consuming energy drinks, the frequency of consumption differed by school type (*p* = 0.002). The proportion of secondary school students drinking these beverages daily or almost daily (18.6%) or 3–4 times per week (8.7%) was higher than that of primary school students (12.8% and 4.7%, respectively). The median frequency of consumption was 1–2 times per month for primary school students and 1–2 times per week for secondary school students.

When asked about their reasons for consuming energy drinks (multiple responses allowed), the most frequently reported reason was taste, cited by 63.3% of students who consumed these beverages. This was significantly more common among girls than boys (70.3% vs. 58.6%; *p* = 0.002). The second most common reason was a desire to increase physical and mental performance (51.2%). Smaller proportions of students reported consuming energy drinks because they were fashionable (5.2%) or to avoid social exclusion from their peer group (5.0%).

### 3.2. Attempts to Purchase Energy Drinks

Over the past year, 34.2% of adolescents surveyed reported attempting to purchase energy drinks in physical shops. Attempts were more frequent among secondary school students (39.4%) than primary school students (17.6%) (*p* < 0.001), among rural residents (42.6%) compared with urban residents (27.3%) (*p* < 0.001) and among boys compared with girls (37.4% vs. 29.8%; *p* = 0.001). When asked whether the retailer requested proof of age, only 19.3% of respondents answered ‘always’ (Figure 3). No statistically significant differences were observed between primary and secondary school students in this regard.

Only 13.8% of students attempting to purchase energy drinks reported being always refused sale. A total of 48.2% stated that sellers sometimes refused the sale, while 38.0% indicated that sellers never refused to sell them these beverages. Secondary school students were less likely than primary school students to encounter refusal. Among those attempting to purchase, 39.9% of secondary school students and 24.3% of primary school students reported never being refused a sale (*p* = 0.033). Girls were also less likely than boys to encounter refusal (*p* = 0.028) (Table 2).

Only 4.5% of students surveyed reported attempting to purchase energy drinks online. No statistically significant differences were observed between primary and secondary school students (3.0% vs. 4.9%), between girls and boys (3.6% vs. 5.1%) or between urban and rural residents (3.9% vs. 5.2%). Among those who attempted to purchase energy drinks online, as many as 71.1% reported being able to complete the purchase independently, 22.4% could do sometimes and only 6.6% were never able to purchase online. A statistically significant difference was observed between primary and secondary school students in the distributions of responses (*p* = 0.043). Secondary school students were more likely to be able to complete the purchase independently always (76.6% vs. 41.7%) and less likely to be able to do so sometimes (18.8% vs. 41.7%) or never (4.7% vs. 16.7%).

When asked whether peers believe that sellers always comply with the ban on the sale of energy drinks to minors, the majority of respondents (61.8%) disagreed. No statistically significant differences were observed between primary and secondary school students in this regard. The majority of respondents (58.1%) reported that the ban on sales to minors does not impede their access to energy drinks, whereas 41.9% believed that the ban prevents access. Among those who stated that the ban did not hinder access, the most frequently reported methods of obtaining energy drinks were: purchasing through adults (generally older friends, but in some cases parents), cited by 35.5% of students and sellers not checking proof of age, cited by 30.0%. Other methods included purchasing with a fake ID (2.0%), buying online (1.7%) or appearing older than one’s age (1.5%). 29.3% of students did not provide an explanation for why the ban was not a barrier to minors obtaining energy drinks. The frequency of these responses was not associated with school type, gender or place of residence.

### 3.3. Perception of the Harmfulness of Energy Drinks

More than half of the students surveyed (57.0%) reported that the ban on the sale of energy drinks to individuals under 18 years of age did not convince them of the harmful effects of these beverages on the health of children and adolescents. The remaining 43.0% indicated that the ban had highlighted their potential harm. Personal drinking habits were clearly linked to the perception of harmful effects. As many as 55.5% of non-drinking students declared they believed energy drinks were harmful, compared to only 25.0% of those who drank (*p* < 0.001). Students unconvinced of the negative impact of energy drinks on health consumed them more than twice as frequently as those who acknowledged their potential harm (54.0% vs. 23.9%; *p* < 0.001). Significant differences were observed between primary and secondary school students, as well as between urban and rural residents. Primary school students were more likely than secondary school students to perceive energy drinks as harmful (57.8% vs. 38.4%; *p* < 0.001), as were students living in urban areas compared with those living in rural areas (46.0% vs. 39.4%; *p* = 0.006).

## 4. Discussion

The study demonstrated that despite the ban on the sale of energy drinks to minors, consumption among adolescents remains common, affecting nearly half of secondary school students. It should be noted that the survey only asked about consumption during the preceding month, suggesting that the overall proportion of students who consume energy drinks less frequently may be even higher. At this point, it is worth citing a recent study on Polish adults’ perceptions of the implemented restrictions on energy drink sales, which found that one-third of respondents reported seeing adolescents consume these beverages in the past month [24].

The literature lacks studies on energy drink consumption among adolescents in other countries where sales restrictions have been in place for many years. To the authors’ knowledge, this is the second such study in Poland. In our study the level of consumption observed in the second year of the ban (41% of surveyed adolescents) indicates only a modest decline in the popularity of these products, which is consistent with the results of the study by Musz et al., conducted in the first months of the sales ban [25]. In their study, over 52% of adolescents aged 15–17 from the Podkarpackie region declared that they still consumed energy drinks, although the majority of consumers stated that they had reduced their consumption. Given the results of both studies, we agree with the authors of the cited study that legislation alone is not enough to effectively limit youth access to energy drinks and that a comprehensive approach is necessary, combined with educational initiatives and effective law enforcement mechanisms. Prior to the introduction of the sales ban, an international study including Poland on adolescent health behaviours (aged 11–15 years) reported that over 46% of Polish adolescents consumed energy drinks [26]. In a study by Nowak and Jasionowski among individuals aged 12–20 years, 53% reported consuming energy drinks in the past month [5], with slightly higher proportions reported by Żyłka and Ocieczek in their study of adolescents aged 13–19 years, where 66% reported consumption in the past month [27]. Notably, the latter two studies also included older adolescents (18–20 years) than those in our sample and as previous research shows, increasing age is associated with more frequent energy drink consumption [9,28,29]. In a Polish study by Granda et al., energy drink consumption was reported by 27% of 10-year-olds, 47% of 12-year-olds and 65% of 14-year-olds [29]. This trend is reflected in our findings, where the risk of consumption increased substantially with each additional year of age. According to global literature, 68% of adolescents aged 10–18 years in Europe consume energy drinks [30] and in the United States, the proportion ranges from 43% to 77% of adolescents [31]. The most recent meta-analysis (2024), including over one million participants of various ages, found that among adolescents aged 12–17 years, the proportion reporting consumption of energy drinks in the preceding month was even lower than in our study, at 34% [32].

Daily consumption of energy drinks has previously been reported in 2–10% of Polish adolescents [5,25,26,27,29,33,34], and the proportion observed in our study (7.3%) indicates that the situation has not improved. This finding aligns closely with the aforementioned meta-analysis, which reported that 8% of adolescents consume energy drinks daily [32]. Scientific evidence, including Polish studies, indicates that boys generally consume energy drinks more frequently than girls [5,16,18,27,35,36,37]; however, in our study, the difference between sexes was small and not statistically significant. The higher prevalence of energy drink consumption among rural adolescents observed in our study may be explained by the fact that this group was, on average, one year older than urban adolescents and that more than 55% attended secondary schools. In contrast, another Polish study found no association between place of residence and energy drink consumption [25,29].

The most frequently reported reason for consuming energy drinks was their taste, cited by over 63% of adolescents, followed by the desire to increase physical and mental performance (51%). Preference for taste as a reason for consumption is unsurprising, as other Polish studies have reported that 53–100% of respondents consumed energy drinks primarily for their flavour [25,29,34]. Also, an Australian study found that 66% of participants cited taste as the main reason [38].

Another concerning finding from the study is that only one in five students reported always being required to show proof of age when attempting to purchase energy drinks and only 14% stated that they were always refused a sale by the seller. The lack of age verification was also reflected in adolescents’ perception of the ease of obtaining these beverages. It is unclear how the situation compares in other countries that have introduced sales restrictions, due to a lack of available data. However, in the context of alcohol and tobacco, both subject to long-standing age restrictions, the findings are not surprising. Studies in Poland indicate that 36% of adolescents aged 13–17 years consume alcohol and 38% smoke cigarettes [39]. Moreover, 39% and 31% of these adolescents, respectively, report purchasing alcohol and cigarettes themselves, which also indicates that age restrictions are often not strictly enforced. Regarding the proportion of students who reported that the main method of obtaining energy drinks is through adults (30%), this is lower than the proportion of adolescents who obtain alcohol from older individuals (42%) but higher than those who receive cigarettes from older individuals (24%) [39].

Online sales have gained popularity in recent years, which, in the case of products intended only for adults, poses a risk of being delivered to minors. Although a small percentage of students in our study attempted to purchase energy drinks online, the majority stated that they could purchase them without any problems. Similarly to sales in brick-and-mortar stores, the scientific literature is lacking in studies on the enforcement of bans on the sale of these drinks online. However, research findings on alcohol sales can again shed light on the issue. A review of international policies governing online alcohol sale and delivery and an analysis of available evidence on retailers’ compliance with these regulations, conducted by Colbert et al., indicate that these regulations may be insufficient to prevent youth access. Therefore, the authors conclude that age verification policies and their practical implementation should be strengthened [40].

A recent Polish study found that over 87% of adult respondents support the ban on the sale of energy drinks to minors, although support increased with age and was lowest among those aged 18–29 years. At the same time, only 45% believed that the ban is effective in limiting adolescents’ access to these beverages [24]. This skeptical attitude is reflected in our study, as more than 58% of adolescents reported that the ban does not pose a barrier to obtaining energy drinks. The study also revealed that the majority of students are not convinced of the harmful effects of energy drinks on the developing body and in practice, these students consumed them significantly more frequently. Although many other studies show that teenagers’ perception of energy drinks influences their consumption levels, it should not be ignored that young people may justify their behavior in this way. In one study, all participants who viewed energy drinks positively reported being consumers, whereas only 26% of those with a negative perception consumed them [27]. In another study of individuals aged 11–13 years who believed energy drinks could be consumed by children, the proportion of drinkers was significantly higher (67%) than among those who disagreed (16%) [41]. In the study by Kwiatkowska et al., the largest proportion of young Poles who did not consume energy drinks cited concerns about their health as the reason for abstaining (49%) [34] and in the study by Mularczyk-Tomczewska et al., 83% of adult Poles considered them as harmful [24].

When interpreting the results of our study, it should be noted that it was not conducted on a nationally representative sample of adolescents. However, the study did include youth from three voivodeships, encompassing schools in the capital city as well as in smaller towns and students residing in both urban and rural areas. Its implementation required approval from school administrations and written parental consent, which, given the associated logistical challenges, makes the study’s scope and sample size satisfactory. It was not possible, however, to ensure an equal proportion of primary and secondary school students living in urban and rural areas. Consequently, 87.2% of primary school students and 44.6% of secondary school students lived in cities, meaning that urban respondents were younger (median age 15 vs. 16 years), which may have influenced the results, particularly regarding urban-rural differences. Given that student recruitment was based on parental consent, it is possible that adolescents whose parents are more liberal towards energy drinks may have been underrepresented. Data from store audits on retailers’ compliance with the sales ban would also be valuable for comparing results.

To the best of the authors’ knowledge, this is the one of the first studies of this kind conducted following the regulation of energy drink sales in Poland. As with any survey-based research relying on memory and self-reported behaviour, adolescents may, consciously or unconsciously, not always provide accurate responses, which should also be considered as a significant limitation of the study. Furthermore, this was a cross-sectional study without the possibility of comparing baseline data.

## 5. Conclusions

The study suggests that the introduction of the ban on the sale of energy drinks to minors in Poland has had only a limited effect on reducing the proportion of adolescents consuming these beverages in the second year of the regulation. However, it should be noted that the study was cross-sectional in nature and was not conducted on a nationally representative sample. Therefore, it is too early to draw general conclusions regarding the effectiveness of the regulation, and further well-designed research is necessary in the coming years. Adults most frequently facilitate access to energy drinks and according to the adolescents, sellers rarely request proof of age when purchases are attempted. Since the sales ban has not convinced the majority of students that energy drinks negatively affect their health, educational campaigns addressing this issue should be implemented.

## Figures and Tables

**Figure 1 nutrients-17-03167-f001:**
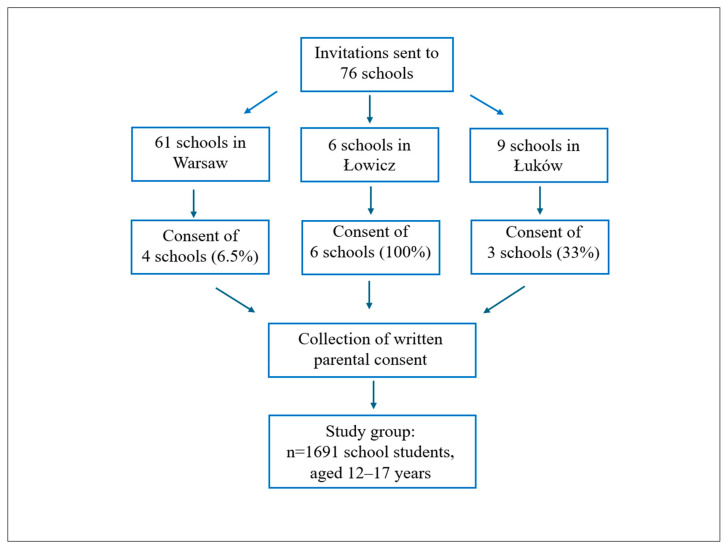
Participant recruitment process.

**Figure 2 nutrients-17-03167-f002:**
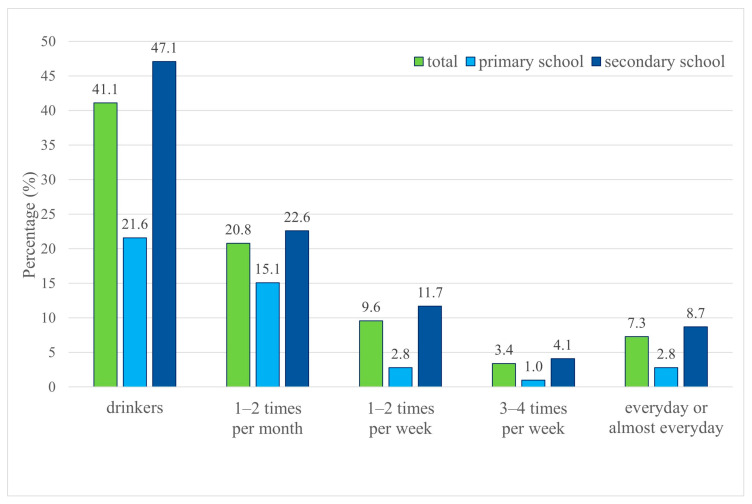
Frequency of energy drinks consumption by school students.

**Figure 3 nutrients-17-03167-f003:**
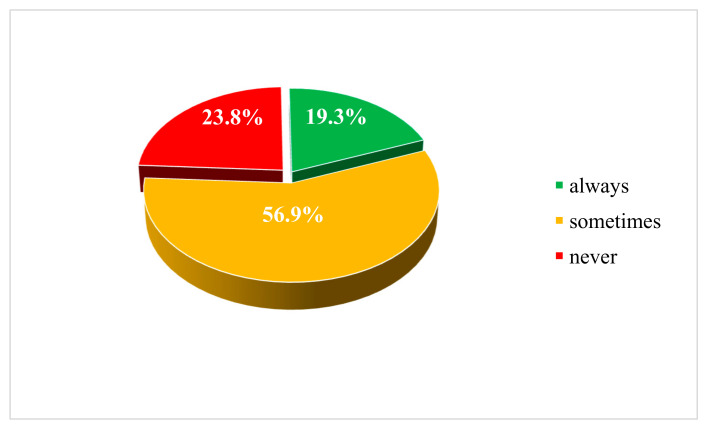
Frequency of age verification by sellers for students attempting to purchase energy drinks.

**Table 1 nutrients-17-03167-t001:** Characteristics of adolescents reporting frequent energy drink consumption (daily, almost daily or 3–4 times per week).

Characteristic	Frequent Consumption (n = 181)	Less Frequent Consumption (n = 1510)	Statistical Significance of Differences (*p*-Value) *
Age (years):			*p* < 0.001 **
range:	13–17	12–17	
quartiles Q1/Q2/Q3	16/16/17	14/16/17	
mean age	16.1	15.5	
standard deviation	1.1	1.4	
School type:			*p* < 0.001
primary	8.3%	25.4%	
secondary	91.7%	74.6%	
Gender:			NS
boys	59.1%	58.5%	
girls	40.3%	41.4%	
non-binary	0.6%	0.1%	
Place of residence:			*p* < 0.001
urban	31.5%	57.4%	
rural	68.5%,	42.6%,	
Awareness of the ban:			*p* < 0.001
yes	88.4%	98.4%	
no	11.6%	1.6%	
Perception that the ban makes obtaining drinks difficult:			NS
yes	37.6%	42.4%	
no	62.4%	57.6%	
Belief in harmful effects:			*p* < 0.001
yes	14.9%	46.4%	
no	85.1%	53.6%	

* Chi-square test, except for comparisons of age distributions. ** Mann–Whitney U test. NS – not significant.

**Table 2 nutrients-17-03167-t002:** Distribution of responses regarding refusal of sale among students attempting to purchase energy drinks in physical shops.

Variables	Refusal Always n (%)	Refusal Sometimes n (%)	Refusal Never n (%)	Total n (%)	Statistical Significance of Differences (*p*-Value) *
Primary school	10 (14.3)	43 (61.4)	17 (24.3)	70 (100)	*p* = 0.033
Secondary school	70 (13.8)	236 (46.4)	203 (39.9)	509 (100)	
Boys	60 (16.2)	182 (49.1)	129 (34.8)	371 (100)	*p* = 0.028
Girls	20 (9.6)	97 (46.6)	91 (43.8)	208 (100)	
Students living in urban areas	37 (14.7)	133 (52.8)	82 (32.5)	252 (100)	NS
Students living in rural areas	43 (13.1)	146 (44.6)	138 (42.2)	327 (100)	

* Chi-square test.

## Data Availability

The data supporting the results of this study will be available from interested researchers upon request. If the data cannot be made publicly available in a trusted repository, the reason for this will be specified in the Data Availability Statement. Further information and materials necessary for the reproduction of the analyses can be obtained from the authors.

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
