# Peer review of "Consumption of Energy Drinks and Attitudes Among School Students Following the Ban on Sales to Minors in Poland"

_nutrients, 2025, doi:10.3390/nu17193167_

Round 1
Reviewer 1 Report
Comments and Suggestions for Authors
Dear authors,
Your manuscript tittled: Consumption of energy drinks and attitudes among school students following the ban on sales to minors in Poland, presents a very interesting nutrition research about energy drink habits in students. Your important study works a serious problem of Public Health in child and young population.
For this reason you have to improve the information in Methodology Part, in particular and the most important aspect is to improve the methodology. You should include information about the Questionnaire, references, validation, implementation, and its use.All of which improves the understanding of your papers and research.
Sincerely yours.
Author Response
Your manuscript tittled: Consumption of energy drinks and attitudes among school students following the ban on sales to minors in Poland, presents a very interesting nutrition research about energy drink habits in students. Your important study works a serious problem of Public Health in child and young population.
For this reason you have to improve the information in Methodology Part, in particular and the most important aspect is to improve the methodology. You should include information about the Questionnaire, references, validation, implementation, and its use. All of which improves the understanding of your papers and research.
Information regarding the research methodology has been added.
Reviewer 2 Report
Comments and Suggestions for Authors
The topic is very interesting and relevant, in terms of health and economy also. The knowledge of harmful components in many premaid foods is long lasting already, but changes are slow, especially in the diet of children and adolescents. The ban of ED selling for younger ages is one important alternative for dealing with this problem. To study the effects of these bans is crucial in many aspects.
The manuscript is well written, fluent, having good questions.
The introduction of results is well structured.
There are very few typos in the text.
There are only few suggestions:
In the Introduction Authors describe the growth of consumption of EDs in Poland and the USA. It would be worth mentioning Europe also, since ED consumption in Europe is later mentioned in the Discussion.
Table 1 in my pdf was in two parts, I guess it won’t happen in the final version. However in Table 1 in the first column the “Age completed” part should be described in more detail under the table (subheading, or short text), because in case readers do not study the full text of the manuscript won’t understand the table. The same regards to the description of “other students” column. Tables, and Figs legends need to be improved.
Reference of Fig3 in the text is after the real figure, it is very difficult to understand the figure that way. A more detailed explanation of it would help.
Most of the Respondents did not drink EDs. What was their opinion regarding the question if Eds are healthy? These data are not evident from the text.
Although referred data and manuscripts are relevant I was finding some papers in PubMed at first glance, and some of those could improve the manuscript:
Med Sci Monit. 2025 Mar 27:31:e947124.
doi: 10.12659/MSM.947124.
Regulatory Efforts and Health Implications of Energy Drink Consumption by Minors in Poland
Paulina Mularczyk-Tomczewska 1, Mariusz Gujski 1, Tytus Koweszko 2, Agata Szulc 3, Andrzej Silczuk 2
Nutrients. 2025 Aug 20;17(16):2689.
doi: 10.3390/nu17162689.
The Frequency, Preferences, and Determinants of Energy Drink Consumption Among Young Polish People After the Introduction of the Ban on Sales to Minors
Patrycja Musz 1, Wiktoria Smorąg 1, Gabriela Ryś 1, Krzysztof Gargasz 2, Ewelina Polak-Szczybyło 3
Dev Period Med. 2016 Apr-Jun;20(2):150-6.
Caffeine intake from carbonated beverages among primary school-age children
Regina Wierzejska 1, Katarzyna Wolnicka 2, Mirosław Jarosz 2, Joanna Jaczewska-Schuetz 2, Anna Taraszewska 2, Magdalena Siuba-Strzelińska 2
Addiction. 2024 Mar;119(3):438-463.
doi: 10.1111/add.16390. Epub 2023 Nov 15.
Prevalence of energy drink consumption world-wide: A systematic review and meta-analysis
Gema Aonso-Diego 1, Andrea Krotter 1, Ángel García-Pérez 2
- PMID: 37967848
- DOI: 1111/add.16390
J Am Heart Assoc. 2019 Jun 4;8(11):e011318.
doi: 10.1161/JAHA.118.011318. Epub 2019 May 29.
Impact of High Volume Energy Drink Consumption on Electrocardiographic and Blood Pressure Parameters: A Randomized Trial
Sachin A Shah 1, Andy H Szeto 2, Raechel Farewell 2, Allen Shek 1, Dorothy Fan 2, Kathy N Quach 2, Mouchumi Bhattacharyya 3, Jasmine Elmiari 2, Winny Chan 2, Kate O'Dell 1, Nancy Nguyen 1, Tracey J McGaughey 4, Javed M Nasir 5, Sanjay Kaul 6 7
Nutrients. 2023 May 29;15(11):2537.
doi: 10.3390/nu15112537.
Energy Drinks and Adverse Health Events in Children and Adolescents: A Literature Review
Pengzhu Li 1, Nikolaus Alexander Haas 1, Robert Dalla-Pozza 1, André Jakob 1, Felix Sebastian Oberhoffer 1, Guido Mandilaras 1
Author Response
The topic is very interesting and relevant, in terms of health and economy also. The knowledge of harmful components in many premaid foods is long lasting already, but changes are slow, especially in the diet of children and adolescents. The ban of ED selling for younger ages is one important alternative for dealing with this problem. To study the effects of these bans is crucial in many aspects.
The manuscript is well written, fluent, having good questions.
The introduction of results is well structured.
There are very few typos in the text.
This has been corrected.
There are only few suggestions:
In the Introduction Authors describe the growth of consumption of EDs in Poland and the USA. It would be worth mentioning Europe also, since ED consumption in Europe is later mentioned in the Discussion.
In the first sentences, the authors emphasize the increase in energy drink sales. Regarding the consumption of these beverages in Europe, relevant information has been added.
Table 1 in my pdf was in two parts, I guess it won’t happen in the final version. However in Table 1 in the first column the “Age completed” part should be described in more detail under the table (subheading, or short text), because in case readers do not study the full text of the manuscript won’t understand the table. The same regards to the description of “other students” column. Tables, and Figs legends need to be improved.
This has been corrected.
Reference of Fig 3 in the text is after the real figure, it is very difficult to understand the figure that way. A more detailed explanation of it would help.
In the manuscript we submitted, the reference to the figure is above the figure; this may be a technical issue.
Most of the Respondents did not drink EDs. What was their opinion regarding the question if Eds are healthy? These data are not evident from the text.
This data has been supplemented in the text.
Although referred data and manuscripts are relevant I was finding some papers in PubMed at first glance, and some of those could improve the manuscript:
Some of the articles you mention are already cited in the manuscript. One article on energy drink consumption following the introduction of the sales ban in Poland, published after our manuscript was submitted to the journal's editorial office, has also been added.
Reviewer 3 Report
Comments and Suggestions for Authors
It was a pleasure to read you paper. However there is still work to do, to reach a high quality paper.
In the document I send some details.

Author Response
This manuscript addresses an important and timely public health issue—the impact of
Poland's 2024 ban on sales of energy drinks to individuals under 18 years of age. The
authors present valuable, survey-based data from over 1,600 adolescents concerning
their consumption patterns, their perceptions of the drinks' potential harm, and their
experiences with age verification. While the study offers relevant insights, particularly
given the lack of data since the regulation took effect, several methodological and
interpretive issues must be addressed before the work can be considered for publication
in Nutrients.
The evaluation of a newly implemented national regulation is important for two reasons.
First, it contributes to national discussions about the effectiveness of legislative
measures to curb adolescent energy drink consumption. Second, it contributes to
international discussions about these measures.
The study included nearly 1,700 adolescents from three provinces, spanning both urban
and rural areas. This breadth strengthens the dataset compared to smaller-scale Polish
studies.
Using odds ratios, confidence intervals, and subgroup comparisons provides
transparency and robustness.
The finding that adult-mediated access and lax retailer enforcement undermine the law
highlights actionable policy gaps.
However, in my opinion, several points should be improved in order to better
demonstrate the study's importance and value. As follows:
First, the study is cross-sectional and limited to three provinces. While diverse, it cannot
be considered nationally representative. The lack of stratification by socioeconomic
status further limits its generalizability. The authors acknowledge this, but they should
discuss the implications for interpreting national trends more explicitly.
Recruitment relied on school participation and parental consent, which may introduce
selection bias. Adolescents whose parents are more permissive about energy drinks
might be underrepresented.
These issues have been added in the section regarding the limitations of the study.
Data were collected only in the second year following the ban. It is too early to draw
conclusions about the long-term effectiveness of the policy, as changes in availability,
enforcement, and adolescent culture may take years to manifest. A longitudinal follow-
up or trend analysis would provide more insight.
The authors agree with this view. We concluded in the text that a small impact of the sales ban was observed in the second year of the regulation, but these issues have now been further expanded.
All data on consumption, purchase attempts, and perceptions are self-reported. Due to
recall bias or social desirability, adolescents may underreport or overreport. This should
be acknowledged more explicitly as a limitation.
This information has been highlighted more prominently.
The study equates "effectiveness" primarily with reported consumption in the past
month. However, effectiveness could also be evaluated by comparing trends in
frequency, quantity, or patterns of access with pre-ban benchmarks. The comparison to
older studies is indirect, involving different methodologies and age ranges, which makes
conclusions about effectiveness tenuous.
We have added an important article on energy drink consumption by teenagers following the introduction of the sales ban in Poland, which was published after our manuscript was submitted to the journal's editorial office. At the same time, we understand this point of view, however, there are no two identical studies in the literature on the same topic. Each study has its own time frame and methodology and almost never includes the same individuals, which means that the conclusions are always subject to some degree of uncertainty.
The discussion indicates that many adolescents do not believe that energy drinks are
harmful, but it cannot be determined whether this belief is associated with their
consumption. Those who consume more may downplay the harm to justify their
behavior. A more cautious interpretation is warranted.
Like our study, many other studies cited in the discussion show that people who claim that energy drinks are not harmful are more likely to consume them. Of course, we agree that this may explain their behavior, but such a correlation cannot be precisely determined. With this in mind, we decided to add relevant information on this topic to the article.
A deeper engagement with comparative policy literature would be beneficial for the
manuscript. For instance, it mentions that Sweden, Lithuania, and the UK have
restrictions but offers little critical discussion of compliance mechanisms, enforcement
strategies, or cultural differences that could impact outcomes.
Unfortunately, there are no studies in the scientific literature on the effects of introducing sales restrictions in other countries, so we are unable to discuss differences in enforcement.
The phrase "ban on sales" should be clarified. This is because adolescents still obtain
drinks via adults and online sales. It would be more accurate to describe this as a sales
restriction rather than a complete ban.
In the study, we use wording that is consistent with the law. Although our results suggest poor effectiveness in reducing energy drink consumption, we cannot change this wording to ‘sales restrictions’ because it would not be consistent with the provisions of the law.
While the figures are clear — for example, the frequency of age verification — some
tables include information that is already summarized in the text. Streamlining would
improve readability.
In our opinion, the text addresses only the most general issues, which serve as an introduction to the data in the table. It would probably not be optimal for the reader to include only the tables.
Although parental consent was obtained, it would be beneficial to explicitly state
whether school or regional ethics boards provided oversight, even if formal IRB review
was deemed unnecessary.
The study was conducted in schools by school pedagogues, in the presence of classroom teachers, and no external reviewers exercised additional control.
This study is a useful first step in evaluating the impact of Poland's sales restriction on
adolescent energy drink consumption, which is an important issue that should be further
explored. However, its conclusions about "limited effectiveness" should be tempered
given the methodological constraints and short evaluation period. I recommend the
following major revisions:
- Conclusions should be rephrased to highlight preliminary insights rather than
definitive judgments about policy failure.
The conclusions have been modified
- Broaden the conversation about restrictions, especially regarding representativeness,
self-report bias, and how it aligns with previous research.
A more nuanced account of international policy experiences and enforcement
mechanisms is needed.
Information about the limitations of the study has been expanded
Think about emphasizing the importance of longitudinal monitoring, mixed-methods
strategies (e.g., retailer compliance audits), and nationally representative data to
enhance future evaluations
All these issues have been taken into account.
Round 2
Reviewer 1 Report
Comments and Suggestions for Authors
Dear Authors,
I am sorry but in your new Manuscript in particular in Supplementary Informations doesn´t appear the important information about the Questionnarie (Design, Validation, Review, References and a Copy).
Sincerely, It is very important for your research to indicate a Copy or your Questionnarie.
Sencerely yours.
Comments on the Quality of English Language
Author Response
I am sorry but in your new Manuscript in particular in Supplementary Informations doesn´t appear the important information about the Questionnarie (Design, Validation, Review, References and a Copy).
Sincerely, It is very important for your research to indicate a Copy or your Questionnarie.
We sincerely apologize. Due to technical issues during manuscript submission, we accidentally included figures twice instead of copies of the questionnaire. It has now been included as supplementary material. Information about the questionnaire used in the study was previously added to the text.
Reviewer 3 Report
Comments and Suggestions for Authors
Dear Authors
This is a well-conducted and policy-relevant study that provides valuable initial evidence on the impact of Poland’s energy drink sales ban for minors. With revisions addressing methodological clarifications and more cautious interpretation, the manuscript would make a strong contribution to the literature on adolescent health behaviors and regulatory effectiveness.
Some minor concerns that I can address now in this second version:
In terms of the abstract, the conclusion should better reflect the limitations of the study (e.g., cross-sectional design, lack of national representativeness).
The phrase “energy drinks are harmful” may be overly broad. Consider specifying “potential health risks” or “adverse effects.”
Figure 3 (age verification by sellers) could be clarified with percentages labeled directly on the bars.
Some references are local or non-English. Where possible, include comparable international sources to strengthen global relevance.
Some recommendations that I advise the authors to do:
-
Strengthen the limitations section (representativeness, self-reported data, cross-sectional design, lack of baseline comparison).
-
Reframe conclusions to be more cautious regarding the overall effectiveness of the ban.
-
Expand the discussion of online sales and policy enforcement challenges.
-
Consider multilevel modeling or, if not feasible, justify the chosen analytic approach.
-
Revise figures and abstract for clarity and precision.
Author Response
Dear Authors
This is a well-conducted and policy-relevant study that provides valuable initial evidence on the impact of Poland’s energy drink sales ban for minors. With revisions addressing methodological clarifications and more cautious interpretation, the manuscript would make a strong contribution to the literature on adolescent health behaviors and regulatory effectiveness.
Some minor concerns that I can address now in this second version:
In terms of the abstract, the conclusion should better reflect the limitations of the study (e.g., cross-sectional design, lack of national representativeness).
The abstract has been corrected
The phrase “energy drinks are harmful” may be overly broad. Consider specifying “potential health risks” or “adverse effects.”
The suggestion was incorporated and the text was reworded.
Nevertheless, it should be borne in mind that if these drinks were not harmful to children, there would be no need to ban their sale.
Figure 3 (age verification by sellers) could be clarified with percentages labeled directly on the bars.
We are somewhat confused about this comment. The percentages are in the chart area.
Moreover, your first review stated, "The figures are clear—for example, the frequency of age verification."
We don't understand what we should change in the chart.
Some references are local or non-English. Where possible, include comparable international sources to strengthen global relevance.
We excluded some local publications, but some must be retained because they refer to Polish research, even though they were published in Polish. The overwhelming number of cited publications are studies published in English in Western journals.
Some recommendations that I advise the authors to do:
Strengthen the limitations section (representativeness, self-reported data, cross-sectional design, lack of baseline comparison).
Relevant information was added to the study limitations.
Reframe conclusions to be more cautious regarding the overall effectiveness of the ban.
The conclusions have been modified in both the abstract and the "conclusions" section.
Expand the discussion of online sales and policy enforcement challenges.
Relevant information has been added in the discussion.
Consider multilevel modeling or, if not feasible, justify the chosen analytic approach.
Relevant information has been added in the study methodology.
Revise figures and abstract for clarity and precision.
The abstract has been corrected, but Figure 3 is, in our opinion, clear.